# Childhood Uveitic Glaucoma: Complex Management in a Fragile Population

**Valeria Iannucci** [1],[†], **Priscilla Manni** [1],[*],[†], **Giulia Mecarelli** [1], **Sara Giammaria** [2], **Francesca Giovannetti** [1], **Alessandro Lambiase** [1],[*] **and Alice Bruscolini** [1]

1   Department of Sense Organs, Sapienza University of Rome, 00185 Rome, Italy
2   IRCCS-Fondazione Bietti, 00198 Rome, Italy
*   Correspondence: priscilla.manni@uniroma1.it (P.M.); alessandro.lambiase@uniroma1.it (A.L.)
†   These authors contributed equally to this work.

**Abstract:** Uveitic glaucoma (UG) is a potentially blinding complication of intraocular inflammation and is one of the most common causes of secondary glaucoma in pediatric ophthalmology. Overall management of UG is often challenging and requires a multidisciplinary assessment and careful follow-up. The overlap with steroid-induced glaucoma (SIG) is quite common, as well as the failure of medical and surgical therapy; nevertheless, few recent papers have dealt with this topic. We review the features and the clinical approach to UG in childhood, discussing the treatments available in the pediatric population.

**Keywords:** uveitis and intraocular pressure; pediatric uveitis; uveitic glaucoma; pediatric uveitic glaucoma; steroid and glaucoma; pediatric glaucoma drugs; glaucoma surgery

## 1. Introduction

Childhood glaucoma is a heterogeneous group of disorders often associated with severe visual loss affecting individuals under the age of 16 years old (UK, Europe, UNICEF) or under 18 years old (USA). It affects more than 300,000 children worldwide and accounts for 5% of the causes of blindness in the pediatric population [1,2].

The Childhood Glaucoma Research Network (CGRN) proposed a novel classification system to unify nomenclature in childhood glaucoma [3]. According to this classification, the diagnosis of childhood glaucoma requires the presence of two or more of the following criteria:

- Intraocular pressure (IOP): >21 mmHg;
- Visual field defects consistent with glaucomatous optic neuropathy;
- Progressive myopia or myopic shift with increased axial length (AL);
- Presence of Haab striae or increased corneal diameter;
- Progressive increase in cup-disc ratio (C/D), cup-disc asymmetry of 0.2 or more between both eyes and/or focal rim thinning.

The CGRN classification system identifies primary or secondary forms of childhood glaucoma. Primary glaucoma can be further subdivided into Primary Congenital Glaucoma (PCG) and Juvenile Open-Angle Glaucoma (JOAG).

Secondary glaucoma includes four categories: glaucoma following cataract surgery, glaucoma associated with non-acquired systemic disease or syndrome, glaucoma associated with non-acquired ocular anomalies and glaucoma related to acquired conditions such as ocular trauma or uveitis.

Glaucoma secondary to uveitis (UG) is one of the most common causes of secondary glaucoma in the pediatric population. Despite this, few recent papers have dealt with this topic.

In childhood, anterior uveitis is the prevalent cause of secondary glaucoma [4,5]. Moreover, children with uveitis have a higher risk of developing complications, including glaucoma, compared to adults with uveitis [6].

Whilst pediatric uveitis represents 5–10% of all uveitis cases, visual loss is more prevalent and severe, resulting in no light perception in one-third of patients, according to Kanski et al. [7]. Visual outcomes depend on the anatomical location and duration of the uveitis, age of presentation, management of intraocular inflammation and the prescribed corticosteroid therapy [8].

The British Infantile and Childhood Glaucoma (BIG) eye study shows that 5.3–19% of all childhood glaucoma cases in the United Kingdom are caused by uveitis [9].

Paroli et al., reported in a previous work a 25% prevalence of uveitic glaucoma in children affected by uveitis [10]. In a retrospective analysis of 182 pediatric uveitis patients, secondary glaucoma was reported in 48 patients (26.23%) with female predominance (F:M, 29:19) [4].

Complications involving the eye structures in children uveitis differ from the ones in adult patients uveitis in terms of etiology, prognosis and prevalence [6]. Ocular complications occur in around 76% of children with uveitis. Pediatric uveitis is often misdiagnosed, and this increases the onset of well-known complications, such as cataracts, band keratopathy, glaucoma, and cystoid macular edema but also of specific complications for the pediatric population, such as amblyopia [8]. De Boer et al., reported glaucoma as the second most common complication (19%) after cataracts and one of the most common causes of severe visual impairment in children with uveitis [11]. The therapeutic approach of UG is still challenging due to poor randomized controlled trials (RCTs) in this pediatric population and the lack of universally agreed guidelines.

The aim of this paper is to describe the clinical features of uveitic glaucoma in a fragile population, such as pediatric patients, and to provide an update on etiopathogenesis and the current therapeutic approach.

## 2. Classification and Clinical Features of UG in Childhood

The Standardization of Uveitis Nomenclature (SUN) system [12] classifies uveitis according to the anatomic location of inflammation into anterior (iris and the anterior ciliary body), intermediate (posterior ciliary body and vitreous), posterior (retina and/or choroid), and panuveitis (all structures are affected). In addition, this classification provides information about the disease, such as onset (sudden or insidious), duration [limited (≤3 months) or persistent (>3 months)], and course (acute, recurrent, or chronic).

Here we report the features of pediatric diseases and infections which are more frequently related to uveitic glaucoma, according to the SUN system. Among the anterior uveitis, we reported the Juvenile Idiopathic Arthritis JIA-associated uveitis, the herpetic anterior uveitis and tubulointerstitial Nephritis and Uveitis (TINU).

Juvenile Idiopathic Arthritis (JIA) is now defined as arthritis starting before the age of 16 years and lasting for 6 weeks or more. The disease activity reduces around 9 years of age and peaks around puberty. [13,14]. JIA-associated uveitis is a chronic non-granulomatous anterior uveitis, and in 3–7% of cases, it can precede the diagnosis of arthritis [15]. Unlike other forms of anterior uveitis, JIA-associated uveitis has an insidious onset, either unilateral or, more often, bilateral, and it is usually asymptomatic until sight-threatening complications arise. About three-quarters develop chronic inflammation with a high incidence of complications, such as cataracts, glaucoma, band keratopathy, and persistent cystoid macular edema [16]. The severity and chronicity of JIA-associated uveitis vary markedly. Young age at onset of uveitis, male genders, presence of synechiae at presentation and uveitis occurring before arthritis appear to be the most significant risk factors for a severe course of uveitis and developing complications [5]. A minority have mild, self-limiting uveitis, requiring only short-term topical steroids [17].

Herpetic anterior uveitis is another form potentially responsible for secondary glaucoma in children. Infectious etiologies constitute 13% of all pediatric uveitis [18]. Viruses

implicated are Herpes Simplex Virus (HSV) type 1 and type 2, Varicella Zoster Virus (VZV), and Cytomegalovirus (CMV). They can cause acute unilateral non-granulomatous anterior uveitis, often associated with increased intraocular pressure, but they can also cause granulomatous anterior uveitis in chronic stages. Although rare in children, when the infection involves the retina, Acute Retinal Necrosis (ARN) may occur, causing a devastating reduction in visual acuity.

Tubulointerstitial Nephritis and Uveitis (TINU) is a multisystemic autoimmune disease involving uvea and renal tubules and may be triggered by various factors, such as drugs or infections. Its incidence is highest at 15 years of age, but all age groups can be affected. In TINU, the uveitis is anterior and usually involves both eyes r; however, posterior or intermediate uveitis can occur as well.

Among the intermediate uveitis in childhood, pars planitis (PP), early-onset sarcoidosis (Blau syndrome), and juvenile multiple sclerosis (MS) are worth mentioning [19–21]. PP is an idiopathic type of intermediate uveitis accounting for 5–26.7% of all pediatric uveitis and is a diagnosis of exclusion [21]. Blau Syndrome is a systemic inflammatory granulomatous disease that occurs in children under 5 years. It is characterized by uveitis, arthritis (mostly involving the knee and the wrist), and skin lesions, such as erythema nodosum, skin rash, vasculitis etc. Juvenile multiple sclerosis (MS) is rare in children, with an incidence in the literature ranging from 2.7% to 10.5% of the general MS group [22].

Posterior uveitis includes inflammation of the choroid with or without retinal involvement. In children, as in adults, the most common cause is toxoplasmosis [23,24]. In 70–80% of cases, it appears as unilateral focal necrotizing retinochoroiditis with focal vitritis. The anterior segment might be secondarily involved, presenting as granulomatous anterior uveitis and high intraocular pressure. Ocular toxoplasmosis more commonly results from the reactivation of congenital disease, where new active satellite lesions appear next to an atrophic scar with hyperpigmented borders. The absence of scarring lesions suggests acquired disease. Congenital toxoplasmosis occurs due to the passage of T.gondii by the transplacental route. The transmission risk is highest during late pregnancy, but most severe forms occur during the first trimester with the classic triad of retinochoroiditis, cerebral calcifications and seizures [25,26].

Other forms of posterior uveitis are infrequent in childhood. Vogt–Koyanagi–Harada (VKH) is a multisystem inflammatory autoimmune rare disease affecting the eyes, ears, brain, skin and hair [27,28]. It is more frequent in ethnicities with higher pigmentation, such as Asians, Middle Easterners, Hispanics, and Native Americans. The incidence of VKH in the pediatric population is ethnicity-dependent, varying between 0.5 and 3% of all pediatric uveitis [29]. The diagnosis is usually delayed in children compared to adults, so associated ocular complications, including glaucoma, occur more frequently. Therefore, visual outcomes in VKH are usually worse in the pediatric population [30].

Another entity associated with posterior uveitis in children is Behçet's disease (BD). BD is a multisystemic vasculitis that is less common in pediatric patients than adults, and it is characterized by recurrent mucocutaneous ulcers affecting the oral cavity and the genital area. Pediatric BD onset is around 10–15 years of age, and ocular involvement usually occurs within 2–3 months. It includes bilateral recurrent panuveitis with retinal vasculitis and persisting vitritis. Iridocyclitis, episcleritis, retinitis, retinal hemorrhages, optic nerve edema, and cystoid macular edema can also be found [31].

Data on the relative risk of developing glaucoma in pediatric patients according to the type of uveitis [4] are limited and extremely heterogeneous (Table 1).

**Table 1.** Type of uveitis and prevalence of pediatric uveitis with the corresponding relative risk (RR) of UG. JIA: Juvenile Idiopathic Arthritis; VKH: Vogt–Koyanagi–Harada; CMV: Cytomegalovirus; BD: Behçet's disease.

| Diagnosis | Type of Uveitis | Prevalence (%) | Relative Risk of Glaucoma |
|---|---|---|---|
| JIA | anterior | 26.19 | 2.49 |
| VKH | panuveitis | 3.96 | 2.71 |
| CMV | anterior | 4.76 | 1.46 |
| BD | panuveitis | 4.76 | 1.46 |
| Idiopathic | all | 30.15 | 1.28 |

## 3. Pathogenesis of IOP Dysregulation in Uveitic Glaucoma

The pathogenesis of IOP dysregulation in uveitis is not fully understood yet. Uveitis can compromise the delicate balance between secretion and outflow of aqueous humor; therefore, in patients with uveitis, both ocular hypotony and hypertension can occur.

It has been reported that ocular hypertension develops in up to 46% of uveitic patients and that it is more common in patients with chronic inflammation than in those with acute uveitis [32,33]. Ocular hypotony is less common in uveitis, affecting up to 10% of patients and is more common in young patients, mainly those with JIA uveitis [34]. IOP reduction is frequently due to ciliary body inflammation, resulting in decreased aqueous production [35] and can occur as a complication following glaucoma surgery.

Interestingly, raised IOP in uveitis can be documented in eyes with either closed or open iridocorneal angle. In uveitic glaucoma, open-angle is more common than angle closure [36]. Several pathogenetic mechanisms have been hypothesized to cause IOP rise in both closed- and open-angle uveitic glaucoma.

In the closed angle UG, IOP rise can be secondary to three mechanisms: (1) pupillary block caused by posterior synechiae, (2) presence of peripheral anterior synechiae, and (3) forward rotation of the ciliary body, described in VKH Syndrome [37].

In open-angle UG, the disruption of the blood-aqueous barrier due to inflammation increases the level of proteins in the aqueous humor in uveitic eyes. This great amount of proteins (up to 1308 mg vs. 30–50 mg in normal eyes) can interfere with the aqueous outflow. Peretz et al., characterized the protein pattern of the aqueous humor of patients with and without uveitis. The results were very similar to those of serum proteins, supporting the theory that the increased blood-aqueous barrier permeability may be involved in the pathogenesis of uveitis [38]. Ladas et al., demonstrated a relationship between aqueous humor protein concentration and aqueous humor outflow. Specifically, the outflow was significantly reduced in uveitis with high aqueous humor protein levels and appeared to be normal in active uveitis with low flare levels [39]. In addition, mechanical clogging of the Trabecular Meshwork (TM) can result from the high concentration of inflammatory cells and debris in the anterior chamber [38].

The inflammation of the TM, known as trabeculitis, is also supposed to be involved in the development of uveitic open-angle glaucoma. Trabeculitis is frequently characterized by an acute IOP increase, as in the Fuchs heterochromic uveitis (FHU), herpetic uveitis (HU), and Posner–Schlossman syndrome (PSS) subtypes [39].

Murray et al., collected aqueous samples during cataract surgery from 22 patients with inactive uveitis and 24 subjects without uveitis. Patients with uveitis showed a greater expression of interleukin 2 (IL-2), interferon-gamma (INF-$\gamma$), and tumor necrosis factor-alpha (TNF-$\alpha$), related to T-helper 1 inflammatory response, compared to patients without uveitis [40]. Current evidence shows that the T-helper response is involved in uveitis pathogenesis, and the cytokine response may differ in infectious and non-infectious uveitis [41,42]. However, there is still debate whether a specific cytokine pattern may be linked to a higher risk of developing uveitic glaucoma.

Uveitis is an inflammatory disease, and glucocorticoids (GCs) are considered the first-line treatment. In some patients, steroid responsiveness may play a part in the pathogenesis of uveitic glaucoma. The most accepted definition of steroid responsiveness is a clinically relevant IOP increasing more than 10 mmHg over the baseline, which can lead to glaucomatous optic neuropathy over time [43].

In adults, approximately 62% do not show a significant IOP increase after GCs administration [43,44]. Still, GCs responsiveness has a significant impact in children: about 60% of the pediatric population shows a significant IOP increase [43,45] with a higher risk of Steroid-Induced Glaucoma (SIG) than adults.

GCs induce the expression of genes involved in microstructural changes in the trabecular meshwork. The overexpression of inhibitors of Metalloproteinases (MMPs) increases the deposition of extracellular matrix in the TM, resulting in increased stiffness [43,46]. In addition, GCs stimulate the rearrangement of the cytoskeletal network with the production of cross-linked actin networks (CLANs) that inhibits TM contractility [43,47–49]. Dexamethasone and prednisolone induce IOP elevation more often than other GCs, [43,49–53] and topical route administration is more frequently associated with SIG than other routes [43,54,55]. Ocular hypertension occurs in 3–6 weeks after continuous eye drops administration, and IOP usually returns to the baseline values within 2 weeks after the end of treatment [43,56]. However, IOP elevation can be irreversible after GCs cessation once microstructural damage of the trabecular meshwork occurs [43].

## 4. Medical Management

Correct management of the systemic disease underlying the uveitis is crucial to minimize the recurrence of intraocular inflammation and becomes pivotal in preserving visual function in these patients. The use of systemic steroids and/or immunosuppressive medication in pediatric patients requires multidisciplinary collaboration and careful follow-up to monitor possible side effects. Revising the whole medical management for the systemic disease related to uveitis is beyond the scope of this paper; Table 2 outlines the most relevant therapeutic options in the pediatric population [57].

**Table 2.** Medical options in pediatric uveitis.

| |
|---|
| Glucocorticoids |
| Methotrexate |
| Anti-TNF-alpha monoclonal antibody therapy (adalimumab, infliximab) |
| Anti-interleukin 6 receptor antibody therapy (tocilizumab) |
| A CTLA-4-Ig fusion protein (abatacept) |
| Anti-CD20 monoclonal antibody therapy (rituximab) |
| Janus kinase inhibitors (tofacitinib) |

The medical management of UG in children is complex, and the clinician should consider the etiology, the patient's age at presentation and general health and be aware of the known efficacy and safety profiles of each drug. All the IOP-lowering agents are indicated in UG in adults, except for prostaglandin analogs during active inflammation and pilocarpine, whose miotic effect may increase the risk of posterior synechiae [58].

Nowadays, there is a lack of RCTs regarding the treatment of childhood uveitic glaucoma.

Sacchi et al. [59] identified only five RCTs [60–64] evaluating the efficacy of topical IOP-lowering drugs in patients younger than 18 with glaucoma and ocular hypertension (Table 3).

**Table 3.** Main RCTs for medical treatment in pediatric glaucoma.

| Study | Topical IOP Lowering Drugs | Efficacy in IOP Reduction |
|---|---|---|
| Ott et al. (2005) [60] | Dorzolamide 2% vs. timolol gel 0.25–0.50% | 20–23% vs. 25% |
| Whitson et al. (2008) [61] | Brinzolamide 1% vs. levobetaxolol 0.5% | 20% vs. 16% |
| Plager et al. (2009) [62] | Betaxolol 0.25% vs. timolol gel 0.25–0.50% | 9% vs. 12.15% |
| Maeda-Chubachi et al. (2011) [63] | Latanoprost 0.005% vs. timolol 0.25–0.50% | 26% vs. 21% |
| Dixon et al. (2017) [64] | Travoprost with 0.5% vs. timolol 0.25–0.50% | 27% vs. 25% |

However, most patients were diagnosed with PCG and JOAG, and secondary glaucoma other than UG was included. Therefore, no specific conclusions for medical therapy of UG can be drawn from the primary RCTs on pediatric glaucoma.

Even if most glaucoma eye drops are not licensed for children, they currently have an off-label use in pediatric glaucoma, with a good safety profile [65].

Beta-blockers (BBs) and Carbonic Anhydrase Inhibitors (CAIs) are considered first-line options in UG [58]. Topical BBs decrease aqueous production, reducing IOP by approximately 20–25% in adults and up to 36% in children [59].

Effectiveness in IOP lowering is similar among the different beta-blockers agents. Timolol and Betaxolol are available in eye drops, in 0.25% and 0.50% concentrations, with twice daily administration. Timolol is also available in a 0.25% or 0.50% gel with a single-day indication. BBs are primarily prescribed for pediatric glaucoma, but a careful anamnesis is pivotal to prevent systemic side effects in children at risk of hypotension, bradycardia, bronchospasm, and apnoea [65]. In the previously reported RCTs, one patient developed bradycardia, and another had pneumonia as a severe side effect after timolol administration. Cardioselective B1 blockers, such as Betaxolol, are less likely to cause respiratory side effects.

CAIs, such as Dorzolamide or Brinzolamide, decrease aqueous production and reduce the IOP by approximately 20% in adults and 23% in children [59]. CAIs eye drops three times a day in monotherapy seem more effective in children than in adults [59]. Therefore, Dorzolamide or Brinzolamide could be an excellent pharmacological choice in young patients. However, a history of allergy to sulfonamides should be excluded before CAIs prescription because of the possible cross-reactivity between the two classes of drugs.

Corneal edema, superficial punctate keratitis and stinging have been reported as side effects. Systemic adverse events observed in oral administration, such as renal failure, metabolic acidosis or aplastic anemia, are uncommon in topical treatment [65].

Although Prostaglandin Analogs (PgAs) are considered the first-line treatment for non-uveitic glaucoma, their use is not recommended in uveitic eyes because of their pro-inflammatory effect. However, they can be safely used in quiescent uveitis undergoing immunomodulatory therapy. PgAs increase the uveoscleral outflow and are the most effective pharmaceutical option in monotherapy, with IOP reduction of up to 35% in adults and up to 27% in children [59]. They have the best systemic safety profile and show mainly local side effects such as ocular irritation, itchiness, iris and eyelid pigmentation, periocular fat atrophy and eyelash elongation [58]. In addition, reactivation of herpes keratitis, exacerbation of uveitis and macular edema have been reported [66,67]. Latanoprost and Travoprost administered once a day shows an excellent efficacy and safety profile in pediatric clinical trials compared to Timolol or Dorzolamide. However, the IOP-lowering effect seems to be poorer in children than in adults [59].

A phase III trial designed to assess the safety and efficacy of Bimatoprost compared to Timolol in children was prematurely discontinued in 2015 due to insufficient enrollment. (Clinicaltrial.gov identifier: NCT01068964)

Alpha Adrenergic Agonists (AAAs), such as Brimonidine tartrate, are scarcely studied in children. These drugs can activate alpha-2 receptors in the bulbar vasomotor center resulting in life-threatening side effects in patients under 23 months, such as apnoea, lethargy, hypotension, bradycardia and hypothermia. These severe adverse events may be related to an immature and permeable blood-brain barrier, so AAAs are not safely prescribed for children under 2 years old and should be used with caution in older children [65].

## 5. Surgical Management

Surgery is a viable therapeutic option when glaucoma progresses, regardless of maximal medical therapy. However, appropriate planning of glaucoma surgery in pediatric patients is still challenging compared to adults [68,69] because of the poor collaboration affecting the results of examinations, including visual field testing and IOP measurement.

In pediatric UG, there is an intensive wound-healing response due to young age and uveitic inflammation [70] that can affect surgical outcomes. Furthermore, in young patients, preservation of the conjunctiva is crucial for future filtration surgeries.

Unfortunately, RCTs specifically designed for UG in the pediatric population are not available, and for this reason, surgical treatment is not based on strong evidence [70].

According to available literature, goniotomy may be a good first-line treatment for pediatric UG [71,72] because it is minimally invasive, repeatable and preserves the conjunctiva. In addition, vision-threatening complications are very uncommon after this procedure.

Brenda L. et al., in a retrospective case series, reported that a single goniotomy effectively controls IOP in pediatric uveitic glaucoma, with 50% of success at 10 years; a second goniotomy can raise the rate of success up to 70% in 10 years [71].

Goniotomy requires a clear view of the angle, so eyes with narrow or closed angles, band keratopathy, or corneal edema are unsuitable for this approach. In these conditions, ab externo trabeculotomy is preferred.

Qianqian Wang et al., investigated the role of ab externo trabeculotomy in 28 UG eyes. The Authors reported an overall surgical success rate of 81.8% with one or two procedures. The survival probability was similar to those reported for goniotomy [73]. Moreover, despite the greater conjunctival manipulation, a temporal approach could preserve the superior conjunctiva for future glaucoma surgeries.

Filtration surgery, such as trabeculectomy and Glaucoma Drainage Devices (GDDs), is another option designed to drive aqueous humor from the anterior chamber to an external conjunctival filtering bleb. As previously mentioned, in pediatric UG, the intensive wound-healing response due to young age and chronic inflammation [68] may lead to subconjunctival fibrosis and surgical failure.

Antifibrotic agents, such as Mitomycin C (MMC) or 5-fluorouracil (5FU), have significantly improved the long-term rate of success of filtration surgery, but their use is off-label in ophthalmology [74,75]. Although these drugs are currently used in many pediatric fields (such as oncology, otorhinolaryngology, gastroenterology or ophthalmology) [76–78], neither MMC nor 5 FU is officially licensed for children.

Nowadays, postoperative fibrosis remains a significant problem in glaucoma management, especially in uveitic patients. GDDs' success is less related to sub-conjunctival fibrosis than trabeculectomy, so they are a good first choice in UG [69]; some models are commercially available in smaller sizes, suitable for children (Table 4).

**Table 4.** Main Glaucoma Drainage Devices available for pediatric patients.

| Type | Model | Plate Area (mm$^2$) | Plate Material |
|------|-------|---------------------|----------------|
| **Ahmed® Implant *** | | | |
| Pediatric size | S3 | 96 | polypropylene |
| Pediatric size | FP8 | 96 | silicone |
| Pars plana (Ped) | PC8 | 96 | silicone |
| Pars plana (Ped) | PS3 | 96 | polypropylene |
| **Molteno3 ® Implant **** | | | |
| Pediatric | P1 | 80 | polypropylene |

* valved; ** non valved.

Although smaller models have been specifically designed for children, there is a lack of studies comparing the effectiveness of pediatric vs. adult models in the pediatric population. The little available literature does not seem to demonstrate the superiority of one model over the other [79,80]

Interestingly, Hye Jin Kwon et al., reported similar outcomes in GDD and trabeculectomy with anti-fibrotic agents at 5 years in 82 uveitic eyes in adults. However, trabeculectomy was more likely to fail when reactivation of uveitis occurred because of subconjunctival fibrosis [81].

Unfortunately, there is a lack of studies comparing the outcomes of trabeculectomy and GDDs in uveitic children. The surgeon should consider the high risk of trabeculectomy failure in these patients: the overall success rate at $\geq$5 years ranges from 16% to 73% [82].

GDDs' success rate at $\geq$5 years is higher, ranging from 38% to 89% [82]. Nevertheless, GDDs have considerable long-term complications, such as corneal swelling and tube erosion through the conjunctiva.

Therefore, the choice of filtration surgery should be carefully evaluated according to the clinical situation of the young patient. Pediatric uveitic patients often show an early onset of cataracts due to steroid therapy and prolonged ocular inflammation. Clear vision is pivotal to avoid amblyopia in the pediatric population; therefore, cataract extraction is often necessary for children diagnosed with uveitis to achieve acceptable visual outcomes.

Cataract surgery in UG children is challenging for different reasons. First, posterior synechiae, inflammatory membranes on the anterior capsule, small pupils scarcely responsive to mydriatics and capsular bag instability may be intraoperative pitfalls in uveitic eyes. Furthermore, postoperative complications such as macular edema and epiretinal membrane are more likely to occur after phacoemulsification, and any effort should be made to control ocular inflammation before and after cataract surgery in patients with a history of uveitis [83].

Second, the correct timing of cataract surgery in children is still debated. Multiple factors must be considered prior to surgery, such as the patient's age, risk of amblyopia, and the uni- or bilaterality of the cataract. In addition, in childhood Ugs, the timing of phacoemulsification in relation to glaucoma filtering surgery is also critical: if performed separately, phacoemulsification may increase the risk of failure of glaucoma surgery, and glaucoma surgery may accelerate cataract development [84]. Therefore, pediatric patients must be evaluated on a case-by-case basis, using extreme caution, before the final decision is made. Posterior capsulotomy and anterior vitrectomy are recommended in children, and some Authors suggest a complete vitrectomy for young patients with uveitis and cataract associated with JIA [83]. Hydrophobic acrylic Intraocular Lenses (IOLs) are considered the best option for patients affected by uveitis. However, inflammatory membranes and deposits may occur after phacoemulsification with an IOL implant, especially in patients with sarcoidosis and JIA [83]. IOL implant is not recommended in infants younger than 1 year.

Choosing the IOL power is not easy in pediatric patients because of continuous ocular growth during childhood. An under-correction of the IOL power, according to the age of the patient, with a proper refractive correction, usually achieves acceptable results [85].

Surgical aphakia could be an option in patients with uncertain control of inflammation or capsular bag instability. However, IOL implantation is associated with a lower risk of severe hypotony after surgery [86].

## 6. Conclusions and Special Considerations for Children

Childhood uveitic glaucoma is a potentially blinding condition. Clinical management is challenging for ophthalmologists and requires a multidisciplinary approach. Children have a long life expectancy, so a prompt diagnosis is extremely important, as well as a proper IOP-lowering therapy, in order to reduce disease progression and preserve vision for life. Unfortunately, there is a lack of evidence from RCTs in the pediatric population [59] and especially in children with uveitic glaucoma. Most of the data for UG in the pediatric population were collected from retrospective studies with small sample sizes.

In this review, we aimed to summarize the best available evidence that could be useful for ophthalmologists, pediatricians, rheumatologists and other specialists.

A crucial topic is how to define the pediatric population and how to manage different age groups. The International Conference on Harmonisation/Committee for Medicinal Products for Human Use (ICH/CHMP) guidelines define five subgroups in the pediatric population for clinical trials (Table 5).

**Table 5.** Pediatric population subgroups according to the International Conference on Harmonisation/Committee for Medicinal Products for Human Use (ICH/CHMP).

| Pediatric Subgroup | Age |
| --- | --- |
| Preterm newborn infants | <37 gestation week |
| Term newborn infants | 0–27 days |
| Infants and toddlers | 28 days–23 months |
| Children | 2–11 years |
| Adolescents | 12 to 16–18 years (depending on region) |

This heterogeneous definition makes it difficult to provide general results in clinical trials that could be valid for different ages.

Medical therapy is usually the first line of treatment in glaucoma, but we should remember that children are more prone to systemic side effects than adults because of their smaller body mass and blood volume and the immaturity of the brain-blood barrier [65].

Any effort to decrease the systemic absorption of topical treatments is crucial in these young patients. Training parents and caregivers to occlude the lacrimal duct for 2–3 min after eye drop instillation could be useful in long-term medical management. Furthermore, poor compliance with therapy may be more common in the pediatric population, and the resulting suboptimal IOP control increases the risk of poor visual outcomes in glaucoma.

Fixed combinations, possibly in preservative-free formulation, could be a good choice to reduce the number of daily administrations and improve tolerability and compliance. Currently, none of the fixed anti-glaucoma combinations is licensed for children [65].

Anterior chamber implants could be a new therapeutic strategy to overcome compliance issues in children, preventing ocular surface disease induced by topical therapy and preserving healthy conjunctiva for future surgeries. Durysta® (AbbVie, an Allergan company) is a Bimatoprost anterior chamber implant, the only Sustained Release (SR) approved by the Food and Drug Administration (FDA) available for glaucoma treatment in 2022. Two phase 1 studies (Clinicaltrial.gov identifiers: NCT04060758, NCT05333419) are recruiting patients to evaluate the safety, tolerability and effective dose of an SR Latanoprost implant. A phase 1 study (NCT04360174), two phase 2 studies (NCT02371746 NCT05335122), and two phase 3 studies (NCT03868124 NCT03519386) are trying to assess the efficacy of an intraocular Travoprost implant in glaucoma patients. Unfortunately, none of these new

devices has been licensed for children yet, and new clinical trials are needed to assess safety and efficacy in the pediatric population.

A valid therapeutic option for UG could be Rho Kinases Inhibitors (RKIs), such as Netarsudil and Ripasudil, a new pharmacological option in glaucoma treatment. RKIs increase aqueous outflow through the trabecular meshwork, reducing IOP by about 20–25% in adults in monotherapy. Conjunctival hyperemia, conjunctival hemorrhage and cornea verticillata are the most common side effects [58]. Phase I studies show that RKIs are involved in anti-inflammatory molecular pathways, and so they might be helpful in UG management [87]. Moreover, Kusuhara et al., evaluated Ripasudil eye drops administered twice a day as monotherapy or add-on treatment in 21 eyes affected with uveitic glaucoma in a retrospective case series. The Authors reported no reactivation of uveitis, with effectiveness and safety comparable to other glaucoma types [88]. Although no data are available for children yet, in the future, RKIs might be a specific therapeutic option for uveitic glaucoma in the pediatric population.

In current opinion, surgery is considered the mainstay treatment of pediatric glaucoma for different reasons. First, according to the literature, pediatric patients have an increased probability of non-response to medical therapy [59] and of side effects. In our opinion, surgery should also be considered for children not compliant with topical therapy.

Moreover, young patients have a long life expectancies, long disease durations, and are more likely to need multiple surgeries during their life. Therefore the surgeon should choose an approach that ensures effective IOP control over years and minimizes conjunctival scarring to prevent the failure of future surgeries. In addition, pediatric patients may not tolerate postoperative discomfort, so they need careful postoperative management to avoid complications. Available literature does not suggest significant differences in effectiveness between the surgical approaches described above [80], but a stable control of the disease underlying the uveitis is strongly recommended to avoid further reactivation and possible surgical failure [57].

Nowadays, the surgical treatment of pediatric UG has scarce evidence-based guidelines. Most authors strive to collect large datasets for clinical studies, and reliable prospective analysis is often impossible. Therefore the results are unpowered compared to other pediatric glaucoma types. To provide further insight into this particular topic in a fragile population, large, controlled, well-designed, and prospective studies with longer follow-ups are needed.

In conclusion, pediatric UG is a complex disease requiring multidisciplinary management and careful follow-up. Ophthalmologists, pediatricians, and rheumatologists have to work in close tandem to achieve the highest quality of life for patients and their caregivers [89].

**Author Contributions:** Conceptualization: V.I. and P.M.; investigation: V.I., P.M. and G.M.; methodology, A.B. and A.L.; writing—original draft preparation, V.I., P.M. and G.M.; writing—review and editing, S.G., F.G. and A.B.; project administration: A.B.; supervision, A.L. All authors have read and agreed to the published version of the manuscript.

**Funding:** This research received no external funding.

**Institutional Review Board Statement:** Not applicable.

**Informed Consent Statement:** Not applicable.

**Data Availability Statement:** All data analyzed during this study are included in this published article.

**Conflicts of Interest:** The authors declare no conflict of interest.

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
