# Peer review of "Childhood Uveitic Glaucoma: Complex Management in a Fragile Population"

_applsci, doi:10.3390/app13042205_

Round 1

Reviewer 1 Report

Although pediatric glaucoma is less frequently found than adult glaucoma, it should be continuously studied and every ophthalmologist must fully be aware of its up-to-date knowledge and treatment. At that point, I appreciate authors who illuminated and shared that information. However, I found that there was no mention of a comparison between FP7 (Adult) and FP8 (Pediatric)  in the Ahmed valve section. Although FP8 was designed for childhood glaucoma, the outcome is not convincing and promising compared to FP7. (FP8 and FP7 have the same inner and outer diameters, but the surface area is different in the two models. Considering the surface area is a significant contributor for surgical success, these parts should be stated in your article)

There are some articles out in Pubmed, so I suggest that authors go through them and add them to this article. 

Otherwise, it seems to me this article will provide good guidance to ophthalmologists who treat the children population. 

Author Response

Dear reviewer,

please find attached our response to your kind comment.

Reviewer 2 Report

This review paper summarized current findings and treatment on pediatric uveitic glaucoma. It is interesting to know how the disease affects children. The manuscript included a lot of references. However, the way to present all information can be more organized.

1.     The mixed format in the manuscript affects reading experience. Some paragraphs have indentation, while some don’t have. Moreover, there are too many paragraphs in this paper. Some may be accidentally separate from the previous paragraph.

2.     There is redundant information, such as the whole paragraph of JIA in the Classification section. A few sentences could be enough to characterize this systemic disease and focus on the UG caused by it.

3.     The Classification begins with several systemic disease and then change to intermediate and posterior. Due to the combined information, using the anatomic location to classify, or summarize all systemic disease in a subsection would be more organized and friendly to readers.

4.     For Table 1, what is the unit for risk of glaucoma? Is it percentage?

Author Response

Dear reviewer,

please find attached our response to your kind comments.
